# Degradation Attacks on Certifiably Robust Neural Networks

**Klas Leino**[*]                                             *kleino@cs.cmu.edu*
*Carnegie Mellon University*

**Chi Zhang**[*]                                             *chiz5@andrew.cmu.edu*
*Carnegie Mellon University*

**Ravi Mangal**[*]                                             *rmangal@andrew.cmu.edu*
*Carnegie Mellon University*

**Matt Fredrikson**                                             *mfredrik@cmu.edu*
*Carnegie Mellon University*

**Bryan Parno**                                             *parno@cmu.edu*
*Carnegie Mellon University*

**Corina Păsăreanu**                                             *pcorina@andrew.cmu.edu*
*Carnegie Mellon University*

**Reviewed on OpenReview:** *https://openreview.net/forum?id=POXO5ZE98j*

## Abstract

Certifiably robust neural networks protect against adversarial examples by employing run-time defenses that check if the model is certifiably locally robust at the input under evaluation. We show through examples and experiments that any defense (whether complete or incomplete) based on checking local robustness is inherently over-cautious. Specifically, such defenses flag inputs for which local robustness checks fail, but yet that are not adversarial; i.e., they are classified consistently with all *valid* inputs within a distance of $\epsilon$. As a result, while a norm-bounded adversary cannot change the classification of an input, it can use norm-bounded changes to degrade the utility of certifiably robust networks by forcing them to reject otherwise correctly classifiable inputs. We empirically demonstrate the efficacy of such attacks against state-of-the-art certifiable defenses. Our code is available at https://github.com/ravimangal/degradation-attacks.

## 1 Introduction

An adversarial example for a neural classifier is the result of applying small modifications to a correctly classified *valid* input such that the modified input is classified incorrectly. For neural classifiers trained in a standard manner, it has been shown that adversarial examples are rampant (Szegedy et al., 2014; Carlini & Wagner, 2017). While training models in an adversarially-aware manner (Madry et al., 2018; Wong & Kolter, 2018; Leino et al., 2021) can help mitigate the issue to an extent, it does not solve the problem since these models continue to be susceptible to adversarial examples. In other words, adversarially-aware training, even with certified methods, does not yield models that are guaranteed to be free from adversarial examples.

For neural classifiers deployed in safety-critical applications like autonomous vehicles and healthcare diagnostics, the existence of a single adversarial example can lead to disastrous consequences. In light of the limitations of training time interventions, how can we ensure that our models are guaranteed protection from adversarial examples? One naive approach is to assume that all the valid inputs a model will ever encounter are known

---

[*]Equal Contribution

in advance. If we can certify that the model is $\epsilon$-locally robust at each of these inputs, then the model is guaranteed to be free from adversarial examples. However, such an assumption is too strong and is unlikely to be true in practice.

*Certifiably robust classifiers* (Cohen et al., 2019; Leino et al., 2021) offer the most rigorous solution to the problem and provably protect models against adversarial attacks. They work by providing a (provable) robustness certificate for each examined input. These papers only discuss and evaluate the certification procedure on a finite set of (valid) inputs with the implicit assumption that either the whole set of valid inputs is contained in the evaluated set or that the certification obtained on a subset of (valid) inputs at test time generalizes also to unseen inputs at run-time. Both these assumptions may not hold in practice. In order to use such methods in safety-critical situations, we therefore assume that these classifiers use the robustness certificate to construct a certified *run-time* defense. In fact, we confirmed with the authors that these certified defenses are indeed meant to be used at run-time. The defenses work by by checking if the model is $\epsilon$-locally robust at the evaluated input. If the check fails, the model rejects the input as the input can be potentially adversarial. This ensures that the model is only ever used for prediction at inputs where the adversary cannot cause a misclassification. However, a rejection is not free of cost. Every time the model rejects an input, a user of the model has to resort to some other process other than their presumed first choice (the defended model) to make a prediction, reducing model utility. The rate of input rejection is then a natural measure of quality for certifiably robust models but, surprisingly, existing evaluations of such models fail to measure this quantity.

We show that existing certified run-time defenses are overly cautious and susceptible to rejecting non-adversarial inputs. This over-cautiousness is inherent in the design of such defenses, and it is manifested even when the $\epsilon$-local robustness check is exact (i.e., the defense is complete). Not only does this lead to a degradation in model utility because of unnecessary rejections, but it also exposes models to a new line of attack that we refer to as *degradation attacks*. We develop new attacks that are aimed at causing certifiably robust classifiers to reject inputs frequently, and we show that this is a significant problem in practice. On MNIST and CIFAR-10, for state-of-the-art certifiably $\ell_2$ robust models, such as GloRo Nets (Leino et al., 2021) and Randomized Smoothed models (Cohen et al., 2019), our attacks successfully find perturbations that lead to unnecessary rejection on as many as 57% of test inputs *where the model is already known to be robust.* On certified $\ell_\infty$ robust models, namely KW (Wong & Kolter, 2018) models, the attack success rates are even higher. This is particularly distressing given the already considerable computational costs of training and defending certifiably robust models. Although we present our ideas in the context of $\ell_p$ norm-bounded perturbation attacks and current defenses, degradation attacks are a concern for any run-time defense that relies on local robustness checks to thwart perturbation attacks.

As a concrete scenario, consider an autonomous driving system that uses a neural classifier for labeling road signs. It is unsafe for the classifier to misclassify adversarially perturbed road signs, and we want to prevent this at all costs. One simple strategy is for the classifier to always reject inputs and either hand over decision-making to the human driver or bring the vehicle to a standstill (UNECE, 2020). This is perfectly safe behavior, but this model has zero utility. Ideally, we want the model to hand over control (i.e., the run-time defense should raise a flag) only when the perturbed example is actually going to cause misclassification. Our result implies that an adversary can cause the model to hand over control to the human driver even when a perturbed input would not have been misclassified. Although this does not happen for all the inputs (as in our simple example), it happens often enough that the adversary can cause a drastic and unnecessary reduction in model utility.

Degradation attacks are successful because certified run-time defenses do not account for the data manifold. We formalize *valid* inputs by means of a set $M$ which is the support set of the underlying input distribution that characterizes the learning problem. Although the exact description of $M$ is unknown, we know that both the training and test data are in $M$. An incorrectly flagged non-adversarial input is an input at which the local robustness check fails but if the model were not to reject the input, it would be classified *consistently* with all valid inputs within an $\epsilon$ distance. While a norm-bounded adversary cannot change the classification of an input in the presence of a certified run-time defense, it can apply norm-bounded modifications to existing valid inputs (i.e., inputs in $M$) and force the model to reject otherwise correctly and consistently classifiable inputs, thereby degrading the model's utility.

As we discuss in Section 5, defenders against degradation attacks have two options: (1) they can train models using existing adversarial training methods with radius $2\epsilon$ when the adversary is only allowed $\epsilon$-perturbations and then compose these models with an existing $\epsilon$-certified run-time defense; (2) they can develop new certified run-time defenses that account for the fact that an input needs to be consistently labeled only with points in $M$ that are $\epsilon$-close. We evaluate the ramifications of the former option that does not require knowledge of $M$. However, the latter option may not be actionable since the certifiable method would need access to a mathematically precise definition of $M$.

To summarize, the main contributions of our work are as follows: (1) we describe new attacks, referred to as *degradation attacks*, that can force certifiably robust models (with either deterministic or stochastic defense mechanisms) to frequently and unnecessarily reject inputs; (2) we empirically demonstrate the severity of the problem for models using state-of-the-art certified defense mechanisms like GloRo Nets and Randomized Smoothing; (3) we make explicit the set, $M$, from which *valid* inputs are drawn, and this helps us explain that certified run-time defenses, based on checking $\epsilon$-local robustness, are susceptible to degradation attacks, as they enforce local robustness (i.e., consistency of the predicted label with all $\epsilon$-close inputs) instead of consistency just with inputs in $M$ that are $\epsilon$-close; and (4) we discuss two possible defenses against degradation attacks, and evaluate the one based on training models with double the radius to be enforced by the certified run-time defense.

The rest of this paper is organized as follows. In Section 2, we revisit definitions of adversarial robustness and sketch a general approach for certified defenses. In Section 3, we demonstrate how a certified defense can unnecessarily reject inputs; we present degradation attack algorithms to this end, and evaluate their efficacy in Section 4. In Section 5, we sketch two defenses against degradation attacks, and concretely evaluate one of these proposals. In Section 6, we describe the related work. Finally, we conclude in Section 7.

## 2 Adversarial examples and local robustness

A neural classifier $f \in \mathbb{R}^d \to L$ is a function from a $d$-dimensional real vector space, $\mathbb{R}^d$, to a finite set of labels, $L$. The inputs to the classifier are drawn from some distribution with support set $M \subseteq \mathbb{R}^d$. In other words, the training and test sets are comprised of elements from $M$ and their corresponding labels. Though the description of $M$ is not available to us, it plays a key role in formalizing the notion of adversarial examples.

An adversarial example for a neural classifier is the result of applying small modifications to a correctly classified *valid* input such that the modified input is classified incorrectly. Formally, an input is valid if it belongs to $M$. Definition 2.1 below, adapted from Szegedy et al. (2014), formalizes the notion of a targeted adversarial example.

**Definition 2.1** (Targeted adversarial example). Given a neural classifier $f \in \mathbb{R}^d \to L$ and a **valid input** $x \in M \subseteq \mathbb{R}^d$, an input $x'$ is a targeted adversarial example with respect to a target label $l \in L$, an $\ell_p$ metric, and a fixed constant $\epsilon \in \mathbb{R}$ if the solution $x'$ to the following optimization problem is such that $||x' - x||_p \leq \epsilon$:

$$\text{Minimize } ||x' - x||_p \text{ subject to } f(x') = l$$

*Untargeted adversarial examples*, or simply *adversarial examples*, are defined similarly except that the constraint $f(x') = l$ is replaced with $f(x') \neq f(x)$.

The fact that $x$ needs to be a *valid* input, i.e., $x \in M$, for $x'$ to be an adversarial example is only stated informally in (Carlini & Wagner, 2017; Szegedy et al., 2014). We make this requirement explicit and formal, via set $M$; as we shall see, this explicit formulation of $M$ will lead us to the design of our degradation attacks.

Note that we can formulate the following alternate definition of an adversarial example (Definition 2.2).

**Definition 2.2** (Adversarial example (alternate)). Given a neural classifier $f \in \mathbb{R}^d \to L$, an input $x' \in \mathbb{R}^d$ is an adversarial example with respect to an $\ell_p$ metric and a fixed constant $\epsilon \in \mathbb{R}$ if

$$\exists x \in M. \ ||x' - x||_p \leq \epsilon \text{ and } f(x') \neq f(x)$$

---

**Algorithm 2.1:** Prediction with a certified run-time defense

---

**Inputs:** A model $f \in \mathbb{R}^d \to L$, an input $x' \in \mathbb{R}^d$, an attack bound $\epsilon \in \mathbb{R}$, and a distance metric $\ell_p$
**Output:** A prediction $l \in L \cup \{\bot\}$

**1** `PredictWithDefense(`$f$ , $x'$ , $\epsilon$ , $\ell_p$`):`
**2**     **if** $f$ *is* $(\epsilon, \ell_p)-$*locally robust at* $x'$ **then**
**3**         **return** $f(x')$
**4**     **else**
**5**         **return** $\bot$

---

The two definitions agree on the set of inputs that are identified as adversarial examples. Clearly, if $x'$ is an adversarial example as per Definition 2.2, then it is also an adversarial example with respect to $x$ as per the untargeted version of Definition 2.1, and vice versa.

A classifier is protected from adversarial examples with respect to a valid input $x$ if it is locally robust at $x$. As stated in Definition 2.3, a classifier is locally robust at $x$, if its prediction does not change in an $\epsilon$-ball centered at $x$.

**Definition 2.3** (Local robustness). A neural classifier $f \in \mathbb{R}^d \to L$ is $(\epsilon, \ell_p)$-locally robust at $x \in \mathbb{R}^d$ if,

$$\forall x' \in \mathbb{R}^d. \, ||x' - x||_p \leq \epsilon \implies f(x') = f(x)$$

Using the notion of local robustness allows us to formally state when a model is free of adversarial examples (we say that the model is robust).

**Theorem 2.4.** *A classifier is free from adversarial examples if it is locally robust at **all** points in $M$.*

*Proof.* Let $A := \forall x \in M. \forall x' \in \mathbb{R}^d. ||x' - x||_p \leq \epsilon \implies f(x') = f(x)$. $A$ formally states that classifier $f$ is locally robust at all points in $M$. Let $B := \exists x \in M. \exists x' \in \mathbb{R}^d. ||x' - x||_p \leq \epsilon \wedge f(x') \neq f(x)$. Then, $\neg B$ formally states that the classifier $f$ is free from adversarial examples. Let us assume $B$. Given $A$, assuming $B$ leads to a contradiction. Hence, $A \implies \neg B$. $\qquad\square$

Note that without $M$, we could not formally state the requirement of freedom from adversarial examples. For instance, one could ignore $M$ and define a classifier to be locally robust at *all* inputs (both valid and invalid); however such a requirement could only be satisfied by the constant function, rendering the classifier useless. Thus, when studying adversarial examples and corresponding defenses, it appears to be essential to consider $M$ explicitly.

Practically, since $M$ is not known a priori (and may be infinite), one can not certify a classifier as free from adversarial examples at test time; a run-time check is needed. In this paper, we observe that certified run-time defenses can erroneously flag non-adversarial inputs because they enforce local robustness, i.e., consistency of the predicted label with all $\epsilon$-close inputs, instead of consistency just with inputs in $M$ that are $\epsilon$-close. In other words, run-time checks should be based on Definition 2.2 instead of simply checking it the model is locally robust at the evaluated input.

**Dichotomy between *defending against adversarial examples* and *certifying local robustness*.** To defend against adversarial examples, researchers have proposed methods based on certifying local robustness. Classifiers, when composed with certified run-time defenses based on local robustness checks, are guaranteed protection from adversarial examples. These defenses, briefly surveyed in Section 6, check for local robustness at a given input, regardless of whether the input is valid or not, and reject inputs for which the check fails. Algorithm 2.1 describes this manner in which certified run-time defenses are deployed for protection.

At a first glance, the certifiably robust models may seem to be working as intended; they reject inputs on which the model's prediction is not robust. However, if the goal of the certified defenses is to protect against adversarial examples (rather than simply ensuring local robustness), we argue that such a defense is too

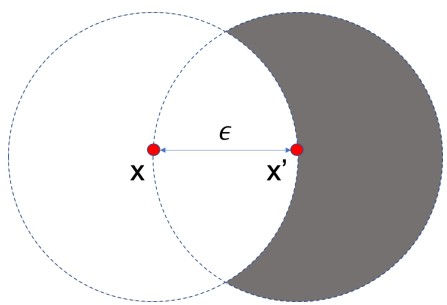

Figure 1: A flagged non-adversarial example

---

**Algorithm 3.1:** Degradation attack algorithm

---

**Inputs:** A model $F \in \mathbb{R}^d \to L$, an input $x \in \underline{D}$, an attack bound $\epsilon \in \mathbb{R}$, and a distance metric $\ell_p$
**Output:** An attack input $x' \in \mathbb{R}^d$

1 DegradationAttack($F$ , $x$ , $\epsilon$ , $\ell_p$):
2     $x' := \text{Attack}(F, x, \epsilon, \ell_p)$
3     **return** $x'$

---

strong, as the rejected inputs may or may not be adversarial, leading to cases when the defense rejects non-adversarial inputs unnecessarily.

Note that even exact (i.e., sound and complete) local robustness certifiers can be overly cautious if deployed at run-time since the model predictions only need to be consistent with valid (i.e., in-distribution) inputs that are $\epsilon$-close and not with all $\epsilon$-close inputs. On the other hand, the argument that classifiers abstain unnecessarily because they ignore the *data manifold* (i.e., $M$) may appear as non-actionable. For a certifiable method to take advantage of the domain of valid inputs, it would need access to a mathematically precise definition of what constitutes a valid input (i.e., the data manifold). It appears that even methods that aim to approximate the *data manifold* cannot be used to actually certify that certain inputs are on the manifold in any meaningful sense. However, we will show in Section 5 that our work *does lead* to concrete actionable interventions.

## 3 Degradation attacks

In this section, we describe our degradation attack algorithms.First, by means of an example, we demonstrate how certified defenses can lead to unnecessary rejections. Next, we describe our degradation attack algorithms against deterministic and stochastic certified defenses.

### 3.1 An example of a certified defense failure

Certified run-time defenses can erroneously flag non-adversarial inputs because they enforce local robustness, i.e., consistency of the predicted label with all $\epsilon$-close inputs, instead of consistency just with inputs in $M$ that are $\epsilon$-close. Consider the example in Figure 1. $x$ is a valid input from $M$. Let us assume that none of the points in the gray region belong to $M$. A binary classifier $f \in \mathbb{R}^2 \to \{W, G\}$ assigns label $W$ to all points in the white region (which includes $x$ and $x'$) and label $G$ to all points in the gray region. $x$ and $x'$ are $\epsilon$ apart and the radius of both the circles is $\epsilon$. The $(\epsilon, l_2)$-local robustness check at $x'$ will fail since the $\epsilon$-ball centered at $x'$ includes white and gray points. However, $x'$ is not an adversarial input since it shares its label with all the $\epsilon$-close inputs in $M$. Consequently, the model rejection of $x'$ is unnecessary and causes a degradation in model utility. This happens even if the local check is precise and the bound $\epsilon$ is tight.

---

**Algorithm 3.2:** Smoothed projected gradient descent attack (SPGD)

---

**Inputs:** A model $f \in \mathbb{R}^d \to \mathbb{R}^{|L|}$ mapping inputs to logit outputs, a loss function $\mathcal{L}$, an input $x \in M$, an attack
        bound $\epsilon \in \mathbb{R}$, a distance metric $\ell_p$, a step size $\eta$, a number of steps $N$, a number of samples $n$, and a
        noise parameter $\sigma$
**Output:** An attack input $x' \in \mathbb{R}^d$

**1** SmoothedPgdAttack($f$ , $\mathcal{L}$ , $x$ , $\epsilon$ , $\ell_p$ , $\eta$ , $N$ , $n$ , $\sigma$):
**2**     $x' := x$
**3**     **for** $0 \leq step < N$ **do**
**4**         $\zeta \sim \mathcal{N}(0, \sigma)^{n \times d}$
**5**         $x' := x' + \nabla_{x'} \mathcal{L} \left( \frac{1}{n} \sum_i f(x' + \zeta_i) \right)$
**6**         $x' := \texttt{Project}(x, x', \epsilon, \ell_p)$
**7**     **return** $x'$

---

## 3.2 Algorithms

**Attacking deterministic defenses.** Algorithm 3.1 describes a simple degradation attack against a deterministic defense. Intuitively, given a valid input $x$ (on which the classifier $f$ is $(\epsilon, \ell_p)$-locally robust, for some fixed $\epsilon$ and distance metric $\ell_p$), we want to find an input $x'$ such that $||x - x'||_p \leq \epsilon$ and the classifier $f$ is not $(\epsilon, \ell_p)$-locally robust at $x'$. The attack directly re-purposes existing white-box attack algorithms (like Carlini-Wagner (Carlini & Wagner, 2017) and PGD (Madry et al., 2018)) that search for adversarial examples. In particular, the attack invokes an off-the-shelf white-box attack algorithm, indicated by `Attack`, to find an adversarial example $x'$ in a ball of radius $\epsilon$ centered at $x$ (line 2). Then, $x'$ is returned as the candidate degradation attack input.

**Attacking stochastic defenses.** Stochastic defenses, such as Randomized Smoothing, transform a base classifier into a smoothed classifier by convolving the base classifier with a Gaussian distribution. The smoothed classifier does not have an explicit representation, and to evaluate it on a single input, the base classifier needs to evaluated on as many as 100,000 samples (Cohen et al., 2019). Moreover, the lack of explicit function representation precludes the use of off-the-shelf white-box attack algorithms that perform gradient descent in the input space. While one could simply perform a standard white-box attack on the base classifier, we found it was more effective to implicitly attack the smoothed model by taking the average gradient over a set of randomly generated samples at each gradient descent step. This gives us a "smoothed PGD" (SPGD) attack tailored for use against stochastic defenses. Our SPGD procedure is given in Algorithm 3.2, and it bears close similarity to the SMOOTHADV$_{\text{PGD}}$ procedure for attacking smoothed classifiers presented by Salman et al. (2019).

Note that Algorithms 3.1 and 3.2 are only meant to serve as proofs-of-concept for degradation attacks. As our empirical results in Section 4.1 demonstrate, even these simple algorithms can be highly effective in degrading model utility.

## 3.3 Threat model

Both the adversary and the defender can draw samples from $M$ (but they may not have full knowledge of $M$). The only assumption in our work is that the adversary can see unperturbed inputs generated by Nature (e.g. images taken with a camera) and these unperturbed inputs are in-distribution samples, i.e., $x \in M$. These are very standard assumptions in machine learning.

The operational details of our attack algorithm are similar to an algorithm for finding an adversarial example in the $2\epsilon$ ball. Technically, our attack is not just finding an adversarial example in the $2\epsilon$ ball. Instead, it tries to find an input $x'$ in the $\epsilon$ ball such that the model cannot be certified $\epsilon$-locally robust at $x'$. However, this similarity to a $2\epsilon$ attack actually reinforces the key point that we are trying to convey. The use of a run-time local robustness check as a defense against adversarial examples allows an $\epsilon$-bounded adversary to

exploit adversarial examples in a $2\epsilon$ ball. To be protected from such degradation attacks, the model is then forced to exhibit $2\epsilon$-local robustness at all points in $M$.

We acknowledge that given the myriad ways, beyond $p$-norm bounded perturbations, in which Nature or an adversary can cause a model to misbehave, our proposed degradation attacks may seem to have a very narrow scope. However, in line with prior works that question existing definitions of adversarial robustness Gilmer et al. (2018); Hendrycks et al. (2021), our work further highlights the issues with $p$-norm based definitions of adversarial robustness as a security property and point to the incompatibility of such definitions with the semantics of the tasks to which machine learning is applied.

For the evaluation (described in the following section), we further make the empirically justified assumption (Yang et al., 2020b) that the data is well-separated, i.e., every pair of valid inputs in $M$ with different ground truth labels are at least $2\epsilon$ apart. Please note that degradation attacks can be successful even when the data is not generally well-separated. We only use this assumption in our experiments for simplifying the calculations and measurements. The adversary does not know or make any well-separateness assumptions (and neither do our attack algorithms). Indeed, since the definition of $M$ is unknown, the adversary does not know if the constructed attack input $x'$ leads to a successful degradation attack or not. It is only during empirical evaluation, where we actually need to know whether the attack was successful or not, that we have to make assumptions about $M$.

## 4    Empirical evaluation of attack efficacy

Our empirical evaluation is designed to measure the susceptibility of state-of-the-art robust models and certified run-time defenses to utility degradation attacks. For our experiments, we consider GloRo Nets (Leino et al., 2021) and Randomized Smoothed models (Cohen et al., 2019). These approaches lead to models with the best known verified robust accuracies (VRA), with respect to the $\ell_2$ metric, on a variety of popular image classification datasets like MNIST (LeCun et al., 2010), CIFAR-10 (Krizhevsky, 2009), and ImageNet (Deng et al., 2009). In Appendix C, we also present an evaluation of degradation attacks on models with certified run-time defenses against $\ell_\infty$ adversaries (in particular, models trained and defended using the convex outer adversarial polytope approach of Wong & Kolter (2018)).

GloRo Nets incorporate global Lipschitz bound computation into the model architecture. The global Lipschitz bound is used to implement the local robustness check at an input under evaluation. Moreover, these local robustness checks are backpropagated over during training to encourage learning of models with a high degree of robustness. In contrast, Randomized Smoothing transforms a base classifier into a smoothed classifier by convolving the base classifier with a Gaussian distribution. The smoothed classifier typically demonstrates a much higher degree of robustness than the base classifier as long as the base classifier has been trained to classify well under Gaussian noise. The method uses statistical techniques to compute the local (certified) robustness radius $R$ of the smoothed classifier at a given input, so an $(\epsilon, \ell_2)$-local robustness check simply amounts to checking if the radius $R$ at the evaluated input is greater than $\epsilon$ or not. Radius $R$ is only probabilistically valid, so all the results for Randomized Smoothing in this section are also probabilistic statements. However, this probabilistic nature of $R$ is typically ignored when metrics like VRA are reported on Randomized Smoothing in the literature, and we do the same here.

We conducted two sets of experiments. In our first set of experiments, described in Section 4.1, we attack certifiably robust models at inputs in the test set using the approaches described in Section 3.2. This demonstrates the efficacy of our attack algorithms, and gives a lower bound on the efficacy of degradation-style attacks in general. Our second set of experiments, described in Section 4.2, measure the susceptibility of models to degradation attacks assuming an all-powerful adversary, i.e., an adversary that succeeds whenever the model is susceptible to a degradation attack at inputs in the test set. This provides an upper bound on the efficacy of degradation attacks.

To measure the efficacy of degradation attacks on a particular model with a certified run-time defense and a dataset, we construct two subsets of the test set for the given dataset, that we refer to as $test_R$ and $test_A$. The former, $test_R$, is the set of test inputs on which the model is accurate and certified to be $(\epsilon, \ell_p)$-locally robust. To construct $test_R$, we simply apply the $(\epsilon, \ell_p)$-local robustness check at every input in the test set

where the model is accurate. If the check passes at an input $x$, it is added to $test_R$. The latter, $test_A$, is a subset of $test_R$, constructed using a degradation attack, such that for each input $x \in test_A$ the model is certified $(\epsilon, \ell_p)$-locally robust at $x$ but there exists an input $x'$ in the $\epsilon$-ball centered at $x$ such that the model cannot be certified $(\epsilon, \ell_p)$-locally robust at $x'$. Since the model is certified locally robust at $x$, we know that $x$ and $x'$ share the same label, so $x'$ is not an adversarial input with respect to $x$ (Definition 2.1). Further, under the assumptions that there exists no other point of a different class label in $M$ that is $2\epsilon$-close to $x$ (Yang et al., 2020b) and for test inputs $x \in test_A$ the model is accurate on all valid inputs (if any) in the $2\epsilon$-ball centered at $x$, $x'$ is not an adversarial example at all (Definition 2.2). Yet, the model would reject $x'$ since the local robustness check fails at $x'$. Thus, under these assumptions, $test_A$ represents the set of test inputs where the model is susceptible to degradation attacks. The ratio $|test_A|/|test_R|$ represents the *false positive rate* of the certified run-time defense, i.e., the rate at which the certified defense erroneously causes the model to reject inputs. We measure model susceptibility and attack efficacy in terms of the false positive rate. A value of one indicates that the model is susceptible to a degradation attack on all test inputs where it is certified locally robust and accurate, while a value of zero indicates that model is safe from degradation attacks on all certifiably robust and accurate test inputs.

We train GloRo Nets using the publicly available code[1]. For Randomized Smoothed models, we use the pre-trained models made available by the authors [2]. The GloRo Nets used in the evaluation are composed of convolution and fully-connected layers. For MNIST ($\epsilon = 0.3$), the model has two convolution layers and two fully-connected layers (2C2F), for MNIST ($\epsilon = 1.58$), the model is 4C3F, and for CIFAR-10 ($\epsilon = 0.141$), the model is 6C2F. For CIFAR-10, the Randomized Smoothed model uses a 110-layer residual network as the base classifier, and for ImageNet, a ResNet-50 model is used as the base classifier We implemented our attacks in Python, using TensorFlow and PyTorch. All our experiments were run on an NVIDIA TITAN RTX GPU with 24 GB of RAM, and a 4.2GHz Intel Core i7-7700K with 32 GB of RAM.

### 4.1 Lower bounds on attack efficacy

To evaluate the efficacy of the attack algorithms presented in Section 3.2 and compute lower bounds on the efficacy of degradation attacks, we construct an under-approximation of $test_A$ (denoted as $\underline{test_A}$) as follows. An input $x \in test_R$ is added to $\underline{test_A}$ if the attack algorithm succeeds in finding an attack input $x'$ in the $\epsilon$-ball at $x$ such that the model cannot be certified robust at $x'$. Table 1 presents lower bounds on degradation attack efficacy against GloRo Nets, measured by performing our degradation attack described in Algorithm 3.1 with PGD used for `Attack`. Table 2 presents lower bounds on degradation attack efficacy against Randomized Smoothing, measured by performing our SPGD degradation attack described in Algorithm 3.2. We employ the same $\epsilon$ values for GloRo Nets as used by Leino et al. (2021), and employ commonly used $\epsilon$ values for Randomized Smoothing that result in visually imperceptible perturbations.

We see that against both GloRo Nets and Randomized Smoothing our attacks decrease model utility to varying degrees. In some cases the reduction in utility was substantial, with false positive rates at 50% or more on MNIST (GloRo Net) and CIFAR-10 (Randomized Smoothing). Overall, false positive rates were higher when the robustness radius was relatively higher; e.g., the false positive rate on MNIST GloRo Nets increases from 6% to 57% when the radius $\epsilon$ increased from 0.3 to 1.58. This may be in part because at smaller radii, many points may be likely to be far more robust than required.

Despite the fact that an explicit representation of smoothed models is not available even to a white-box attacker, we find that our SPGD attack was able to successfully degrade the utility of models defended by Randomized Smoothing. On smoothed models we observe that models trained with a larger noise parameter $\sigma$ are less affected by false positives; however, this comes at the cost of a notable decrease in baseline utility.

In Appendix B.2, we present a visualization of some successful degradation attack inputs against GloRo models, demonstrating that these attack inputs are visually imperceptible from valid inputs.

---

[1] https://github.com/klasleino/gloro
[2] https://github.com/locuslab/smoothing

Table 1: Lower bounds for false positive rates induced by degradation attacks against GloRo Nets. Models are trained and evaluated using the same $\epsilon$ value.

| | baseline VRA (%) | false positive rate (%) | utility reduction (%) |
|---|---|---|---|
| MNIST ($\epsilon = 0.3$) | 95.0 | 6.1 | 5.9 |
| MNIST ($\epsilon = 1.58$) | 61.8 | 57.0 | 34.3 |
| CIFAR-10 ($\epsilon = 0.141$) | 60.0 | 11.2 | 6.6 |

Table 2: Lower bounds for false positive rates induced by degradation attacks against Randomized Smoothing. Similarly to the evaluation of Cohen et al. (2019) results are obtained on a sample of 100 arbitrary test points, as Randomized Smoothing is costly to evaluate.

| | $\sigma$ | baseline VRA (%) | false positive rate (%) | utility reduction (%) |
|---|---|---|---|---|
| CIFAR-10 ($\epsilon = 0.5$) | 0.25 | 48.0 | 50.0 | 24.0 |
| CIFAR-10 ($\epsilon = 0.5$) | 0.50 | 41.0 | 29.3 | 12.0 |
| ImageNet ($\epsilon = 1.0$) | 0.50 | 46.0 | 17.4 | 8.0 |

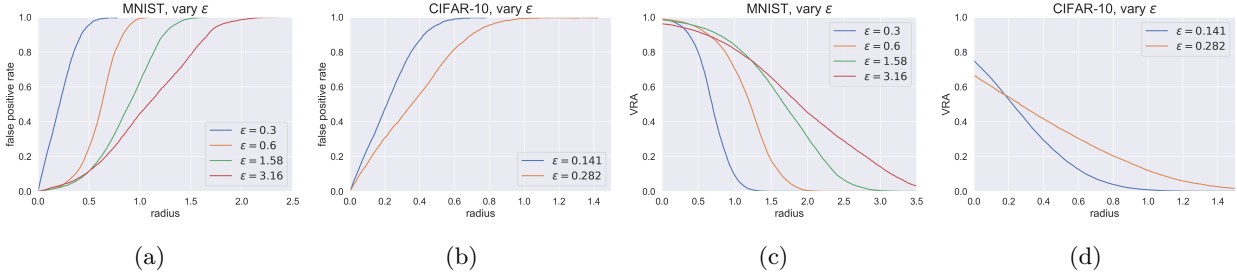

Figure 2: Upper bounds for false positive rates on Gloro Nets

## 4.2 Upper bounds on attack efficacy

To compute upper bounds on the efficacy of degradation attacks, i.e., upper bounds on the false positive rates, we construct an over-approximation of $test_A$ (denoted as $\overline{test_A}$) as follows. We apply a $(2\epsilon, \ell_p)$-local robustness check at every input in $test_R$. If this stronger check passes at an input $x$, it means that no matter how the adversary perturbs $x$ within the $\epsilon$-ball centered at $x$, the model is always $(\epsilon, \ell_p)$-locally robust at the perturbed input, and the certified defense (whether complete or incomplete) cannot be forced to unnecessarily reject the input as long as it satisfies a monotonicity property [3]. GloroNets and Randomized Smoothing are both monotonic in this sense (see Appendix A). However, if the $(2\epsilon, \ell_p)$-local robustness check fails at $x$, then even though the model is certified $(\epsilon, \ell_p)$-locally robust at $x$, there may exist an input $x'$ in the $\epsilon$-ball at $x$ where the model cannot be certified $(\epsilon, \ell_p)$-locally robust and an adversary can force an unnecessary rejection. Therefore, if the stronger check fails at $x$, we add it to $\overline{test_A}$.

Figures 2a and 2b show the false positive rates at different radii (i.e., $\epsilon$ values) for GloRo Net models trained to be robust against the $\epsilon$ values indicated in the legend. The $\epsilon$ values of 0.3 and 1.58 for MNIST and 0.141 for CIFAR-10 are the same as the ones used for evaluating the lower bounds. Figure 2 also reports results for models trained to be robust against twice these $\epsilon$ values. This is for the purpose of evaluating one of our proposed defenses against degradation attacks (see Section 5.1). We see that the false positive rates are quite high and consistently approach one as the radius increases. The models trained with higher $\epsilon$ values are

---

[3]We say that a certification method $\text{cert} : \mathbb{R}^d \times \mathbb{R}^+ \to bool$ is monotonic if, $\forall x, x' \in \mathbb{R}^d, \ \epsilon > 0 \ . \ \text{cert}(x, \epsilon) \ \wedge \ ||x - x'||_p < \epsilon \implies \text{cert}(x', \epsilon - ||x - x'||_p)$.

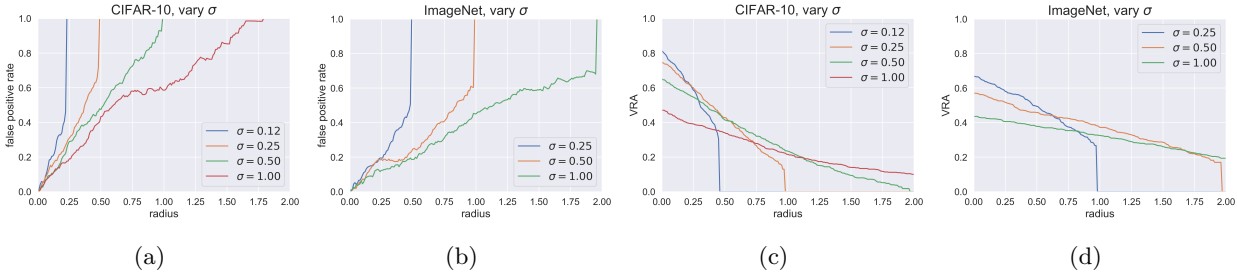

Figure 3: Upper bounds for false positive rates on Randomized Smoothing

more resilient to degradation attacks, but they pay the price of a lower VRA, specially at lower radius values (Figures 2c and 2d). By comparing the upper bounds to the lower bounds obtained in Section 4.1, we see that our degradation attacks recover between $1/4$ and $1/2$ of the best possible degradation.

Interestingly the susceptibility to degradation attacks does not appear to result from the GloRo Net's overapproximation of local robustness. Leino et al. (2021) report that the Lipschitz bounds on the MNIST models are far tighter than on the CIFAR-10 model, yet the CIFAR-10 model is not more susceptible to degradation. As explained in Section 1, this is because the threat of degradation attacks arises intrinsically from any defense that filters based on local robustness at run-time, *not* because of the overapproximate nature of practical defenses.

Figures 3a and 3b show the false positive rates at different radii for each of the Randomized Smoothed models. Again, the false positive rates are high and approach one as $\epsilon$ increases. Models that are smoothed using larger $\sigma$ values unsurprisingly demonstrate more resilience to degradation attacks at the cost of a reduced VRA (Figures 3c and 3d). The sudden jumps in the false positive rate to one indicate the $\epsilon$ value where all points in $test_R$ also end up in $\overline{test_A}$; i.e., the model is susceptible to degradation attacks at all the points where it is certified locally robust.

## 5 Defending against degradation attacks

In this section, we sketch two possible ways of defending against degradation attacks, and empirically evaluate the effectiveness of our first proposal.

### 5.1 Defense via double radius training

A simple strategy to defend against degradation attacks is to first train models using existing adversarial training methods but with radius $2\epsilon$ when the norm-bounded adversary is allowed $\epsilon$-perturbations for adversarial and degradation attacks, and then to compose these models with existing $\epsilon$-certified run-time defenses. This strategy is motivated by the intuition that if a model is $2\epsilon$ robust at a point $x$ in $M$, then no matter how the adversary perturbs $x$ within the $\epsilon$-ball centered at $x$, the model is always $(\epsilon, \ell_p)$-locally robust at the perturbed input, and an $(\epsilon, \ell_p)$-local robustness defense is likely (or even guaranteed, in the case of a complete defense) to not reject the input unnecessarily.

Training a model for $2\epsilon$-robustness is not guaranteed to help because of the frequently observed inverse relationship between accuracy and robustness. The model may be less accurate than the one trained for $\epsilon$-robustness, and may also reject more frequently, even on points in $M$. But validating the model against $2\epsilon$ is certainly helpful. The $2\epsilon$-VRA on validation data is a lower bound on the accuracy one can expect on test data, in the face an $\epsilon$-adversary capable of adversarial as well as degradation attacks. This also suggests that, when evaluating certified defenses, one ought to report not just $\epsilon$-VRA, but also $2\epsilon$-VRA. Validating against $2\epsilon$ can also help in configuring the certified run-time defense. In case the rejection rate is too high at a particular $2\epsilon$, the certified defense can be configured to enforce local robustness at a smaller $\epsilon$ value. This reduces vulnerability to degradation attacks at the cost of an increased vulnerability to adversarial attacks.

We evaluate this defense strategy on GloRo Net models using the same architectures and datasets as in Section 4. However, we re-train all the models using twice the $\epsilon$ values that we want to defend against. Figure 2 summarizes the results. We see that training against $2\epsilon$-robustness helps reduce the false positive rates compared to the models trained against $\epsilon$-robustness. Moreover, this improvement is achieved without significantly affecting the VRA of the models. These results suggest that training against $2\epsilon$-robustness can reduce the vulnerability to degradation attacks, but the false positive rates still remain concerning.

While smoothed models are not explicitly trained for a specific degree of robustness, and therefore this approach does not directly apply, the noise parameter, $\sigma$, does allow one to control the trade-off between robustness and accuracy. Cohen et al. (2019) found that the best VRA is typically achieved when $\sigma \approx \epsilon/2$. Table 2 shows that by increasing $\sigma$ beyond this point, one can obtain better resistance to degradation. However, as Figures 3c and 3d show, increasing $\sigma$ also negatively impacts utility.

### 5.2 Defense via an $M$-membership oracle

Certified run-time defenses which rely on local robustness checks are susceptible to degradation attacks because these defenses are overly strong. They try to enforce local robustness, i.e., consistency of predicted label with all $\epsilon$-close inputs, instead of consistency just with inputs in $M$ that are $\epsilon$-close. To 'fix' these certified defenses, instead of checking local robustness at the given input $x'$, one could instead check if $x'$ shares its label with all the inputs in $M$ that are $\epsilon$-close to $x'$. If the check passes, then $x'$ is guaranteed to be a non-adversarial input, but, more importantly, if the check fails then $x'$ is *guaranteed* to be an adversarial input.

This fix works because an input $x'$ is an adversarial example if and only if there exists some input $x \in M$ that is $\epsilon$-close to $x'$ and is labeled differently from $x'$. To implement our proposed fix, we would need an explicit representation for $M$, or at least an oracle for membership in $M$. A membership oracle may be used to search for members of $M$ that are $\epsilon$-close to a given $x'$. While an exhaustive search for all such members is likely infeasible, we may allocate a fixed budget of computational resources to the oracle-based check. If, within the allocated budget, we find an input $x \in M$ such that $x$ is $\epsilon$-close to $x'$ but it has a different label, then we have detected an adversarial example and can reject $x'$. However, if the check fails to find such an $x$ within the budget, we can fall back to checking local robustness at $x'$. Practically, one could use an Out-of-Distribution (OOD) detector as an $M$-membership oracle. Although OOD detectors can be imprecise and susceptible to adversarial attacks themselves (Sehwag et al., 2019; Bitterwolf et al., 2020), some recent developments (Chen et al., 2020) look promising. We leave this for future work.

## 6 Related work

The discovery of adversarial examples has led to an intense effort in recent years towards formalizing the problem, as well as designing new attack and defense mechanisms.

**Formalizations.** The presence of adversarial examples has been popularly formalized as the absence of local robustness in a neural classifier. Only constant functions are locally robust everywhere, so this definition fails to capture the notion of a classifier free from adversarial examples. Leino et al. (2021) propose the notion of global robustness for classifiers that can reject inputs. They first define an extended notion of local robustness at a point $x$ as the property that every point in the $\epsilon$-ball at $x$ either shares its label with $x$ or is rejected. Then, a classifier is globally robust if it satisfies this notion of local robustness at every input. A different notion of global robustness is proposed by Ruan et al. (2019). They define a model to be globally robust if it is locally robust at every input in a finite test set. These definitions do not capture the idea that, to be protected from adversarial examples, a model only needs to be locally robust at every input in the support set, $M$. For real-world datasets, the support set of the input distribution is much smaller than the entire input space and inputs belonging to different classes are separated by a distance larger than twice the perturbation radii used in adversarial-example experiments (Yang et al., 2020b).

**Adversarial attacks.** Algorithms for constructing adversarial attacks can be divided in to two major categories. White-box attacks assume that they have access to the model internals, i.e., the model architecture

and weights. Their primary strategy is to perform gradient descent in the input space so as to find an input that maximizes the loss while satisfying the constraint of staying within the $\epsilon$-ball centered at the original input. Some popular and successful attack algorithms of this nature include the ones proposed by Goodfellow et al. (2015), Carlini & Wagner (2017), and Madry et al. (2018). Black-box attacks only assume query access to the model (Papernot et al., 2017). In other words, their threat model is weaker than the one assumed by white-box attacks. The degradation attacks proposed in this paper use adversarial attack algorithms as a sub-procedure and are agnostic to whether these algorithms are white-box or black-box.

**Heuristic and certified defenses.** Heuristic defenses against adversarial examples do not provide any guarantees about their defensive capabilities. These include approaches that modify the training objectives, modify the neural classifier post-training, or embed run-time checks to flag adversarial examples during evaluation. The lack of guarantees suggests that these defenses can be broken and it has indeed been demonstrated (Athalye et al., 2018; Tramer et al., 2020) that a number of published defenses are breakable. In this paper, we focus on certified run-time defenses that are deployed during model evaluation and, if an example is adversarial, are guaranteed to flag it (though incomplete defenses can also report false positives). Such defenses check if the classifier is locally robust at the evaluated input. Guaranteed local robustness checks are implemented using a variety of approaches that include constraint solving and formal methods (Huang et al., 2017; Katz et al., 2017; Gehr et al., 2018; Singh et al., 2019), optimization (Bastani et al., 2016; Dvijotham et al., 2018; Raghunathan et al., 2018; Wong & Kolter, 2018; Tjeng et al., 2019), Lipschitz bounds computation (Weng et al., 2018; Leino et al., 2021), and stochastic smoothing (Lecuyer et al., 2019; Cohen et al., 2019; Yang et al., 2020a). Of these, GloroNets (Leino et al., 2021) and stochastic smoothing approaches (Cohen et al., 2019) flag inputs at run-time; we show that run-time defenses based on local robustness checks are overly cautious and often flag even non-adversarial inputs.

The algorithm PREDICT from (Cohen et al., 2019) can return ABSTAIN when the prediction can not be computed for the given input, due to the imprecision of the statistical analysis. This can be potentially exploited by an adversary who can force a certifiably robust classifier to unnecessarily abstain as has been noted in (Cohen et al., 2019). Our attack is different as it is not due to imprecision of statistical tests; it exploits the fact that certified defenses check local robustness and are therefore too conservative, flagging non-adversarial inputs unnecessarily.

## 7 Conclusion

In this paper, we have showed that certified run-time defenses against adversarial examples, based on local robustness checks, are inherently overcautious and can reduce model utility due to unnecessary input rejections. This is a consequence of these defenses enforcing local robustness, i.e., consistency of the predicted label with all $\epsilon$-close inputs, instead of consistency just with *valid* $\epsilon$-close inputs (where valid inputs are those in the support set $M$ of the input distribution). An adversary can exploit this overcautiousness; though it cannot change the classification of a valid input, it can apply norm-bounded modifications to valid inputs and force the model to reject otherwise correctly and consistently classifiable inputs. We have presented concrete degradation attacks of this nature that can be implemented using off-the-shelf white-box attack algorithms. Our empirical evaluation demonstrates that even state-of-the-art certifiably robust models, like Randomized Smoothed models and GloRo Nets, are highly susceptible to utility degradation attacks.

Though the fact that classifiers can abstain unnecessarily because they ignore $M$ may appear as non-actionable due to the lack of a precise definition of $M$, model developers can take the following steps to mitigate degradation attacks:

- Model developers that use certified run-time defenses based on local robustness should evaluate their models with $2\epsilon$-robustness even when the adversary is $\epsilon$-bounded. Verified robust accuracy (VRA) numbers calculated with respect to $\epsilon$-robustness are overly optimistic and do not reflect the performance (% of inputs where the model is both accurate and does not reject) one would see for the deployed model.

- If it turns out that the VRA is too low when evaluated at $2\epsilon$, the developer can configure the certified defense to enforce local robustness at $\epsilon' < \epsilon$. This reduces the vulnerability to degradation attacks at the cost of an increased vulnerability to adversarial attacks.

- As suggested in Section 4.1, the developers can train the model to be robust with respect to $2\epsilon$ perturbations when faced with an $\epsilon$-bounded adversary in order to defend against degradation attacks.

## Acknowledgements

We would like to thank the reviewers for their detailed comments, which helped us improve this article significantly. This material is based upon work supported by Software Engineering Institute under its FFRDC Contract No. FA8702-15-D-0002 with the U.S. Department of Defense, DARPA GARD Contract HR00112020006, a Google Faculty Fellowship, and the Alfred P. Sloan Foundation.

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

# A  Monotonicity of local robustness certifiers

**Definition A.1** (Monotonicity of local robustness certifiers)**.** We say that a certification method `cert` : $\mathbb{R}^d \times \mathbb{R}^+ \to bool$ is monotonic if

$$\forall x, x' \in \mathbb{R}^d, \ \epsilon > 0 \ . \ \texttt{cert}(x, \epsilon) \ \wedge \ ( \ ||x - x'||_p = \delta) < \epsilon \implies \texttt{cert}(x', \epsilon - \delta)$$

**Definition A.2** (Exact local robustness certifier)**.** An exact certification method is one for which $\texttt{cert}(x, \epsilon) \iff \forall x' \in \mathbb{R}^d.||x - x'||_p \leq \epsilon \implies F(x) = F(x')$ (in other words, certification is equivalent to local robustness).

**Lemma A.3.** *All exact certification methods are monotonic.*

*Proof.* This follows essentially from the triangle inequality. Let $x \in \mathbb{R}^d$ and $\epsilon > 0$, and suppose that $\texttt{cert}(x, \epsilon)$. Thus, since `cert` is exact, we have that $\forall x' \in \mathbb{R}^d \ . \ ||x - x'||_p \leq \epsilon \implies F(x) = F(x')$.

Consider $x' \in \mathbb{R}^d$ for which $||x' - x|| \leq \epsilon$, and let $\delta = ||x - x'||$. Now for $x'' \in \mathbb{R}^d$, assume $||x' - x''|| \leq \epsilon - \delta$. Thus,

$$\begin{aligned} \epsilon \ &\geq \ \delta + ||x' - x''|| + \delta \ = \ ||x - x'|| + ||x' - x''|| \\ &\geq \ ||x - x''|| \hspace{4cm} \text{by the triangle inequality.} \end{aligned}$$

Thus, because we have $\texttt{cert}(x, \epsilon)$, we may conclude that $F(x') = F(x) = F(x'')$. This tells us that the classifier $F$ is $(\epsilon - \delta)$-locally robust at $x'$, which, by completion gives us $\texttt{cert}(x', \epsilon - \delta)$. $\qquad\square$

**Definition A.4** (Lipschitz-based local robustness certifier)**.** A Lipschitz-based certification method is one which, given an upper bound, $K_{ij}(x, \epsilon)$ on the local Lipschitz constant of the margin $f_j - f_i$ at $x$,[4] certifies a point $x$, where $F(x) = j$, exactly when

$$\max_{i \neq j} \left\{ f_i(x) + \epsilon K_{ij}(x, \epsilon) \right\} \ \leq \ f_j(x).$$

**Lemma A.5.** *All Lipschitz-based certification methods are monotonic.*

*Proof.* Let $K_{ij}(x, \epsilon)$ be an upper bound on the Lipschitz constant of the function $f_j(x) - f_i(x)$. That is, $K_{ij}(x, \epsilon)$ is the maximum rate of change of $f_j(x) - f_i(x)$ within an $\ell_p$ ball of radius $\epsilon$ centered at $x$.

Let $x \in \mathbb{R}^d$ and $\epsilon > 0$, and suppose that $\texttt{cert}(x, \epsilon)$. Consider $x' \in \mathbb{R}^d$ for which $||x' - x|| \leq \epsilon$, and let $\delta = ||x - x'||$.

We begin with the observation that the $\ell_p$ ball of radius $\epsilon$ centered at $x$—we will denote this as $B(x, \ \epsilon)$—contains $B(x', \ \epsilon - \delta)$. This, too, follows essentially from the triangle inequality:

$$\begin{aligned} z \in B(x', \ \epsilon - \delta) \ &\implies \ ||x' - z|| \leq \epsilon - \delta = \epsilon - ||x - x'|| \\ &\implies \ ||x - x'|| + ||x' - z|| \leq \epsilon \\ &\implies \ ||x - z|| \leq \epsilon \hspace{3cm} \text{by the triangle inequality} \\ &\implies \ z \in B(x, \ \epsilon) \end{aligned}$$

From this, we conclude that Equation 1 holds.

$$K_{ij}(x, \epsilon) \geq K_{ij}(x', \epsilon - \delta) \tag{1}$$

With this in mind, we proceed as follows. Let $j = F(x)$; i.e. $j$ is the class predicted by the classifier at $x$. Since $\texttt{cert}(x, \ \epsilon)$ implies that the classifier is $\epsilon$-locally-robust at $x$, and $x' \in B(x, \epsilon)$, we conclude that $F(x') = j$.

---

[4]note that the global Lipschitz constant is such a bound

---

**Algorithm B.1:** Degradation attack algorithm

---

**Inputs:** A model $f \in \mathbb{R}^d \to L$, an input $x \in M$, an attack bound $\epsilon \in \mathbb{R}$, and a distance metric $\ell_p$
**Output:** An attack input $x' \in \mathbb{R}^d$

```
1 DegradationAttack(f , x , ε , ℓₚ):
2 │   x″ := Attack(f, x, 2ε, ℓₚ)
3 │   x′ := Project(x, x″, ε, ℓₚ)
4 │   return x′
```

---

Table 3: False positive rates induced by degradation attack in Algorithm B.1 against GloRo Nets.

|  | *baseline VRA* *(%)* | *false positive rate* *(%)* | *utility reduction* *(%)* |
|---|---|---|---|
| *MNIST ($\epsilon = 0.3$)* | 95.0 | 6.0 | 5.7 |
| *MNIST ($\epsilon = 1.58$)* | 61.8 | 56.1 | 34.7 |
| *CIFAR-10 ($\epsilon = 0.141$)* | 60.0 | 11.4 | 6.6 |

Because we assume that `cert` is Lipschitz-based (Definition A.4), we have Equation 2.

$$\forall i \neq j \;.\; f_j(x) - f_i(x) \geq \epsilon K_{ij}(x, \epsilon) \tag{2}$$

Since $||x - x'|| = \delta < \epsilon$, by the definition of $K_{ij}$ we have that

$$
\begin{aligned}
\forall i \neq j \;.\; f_j(x') - f_i(x') \;&\geq\; f_j(x) - f_i(x) - \delta K_{ij}(x, \epsilon) \\
&\geq\; \epsilon K_{ij}(x, \epsilon) - \delta K_{ij}(x, \epsilon) = (\epsilon - \delta) K_{ij}(x, \epsilon) \quad\quad \text{by Equation 2} \\
&\geq\; (\epsilon - \delta) K_{ij}(x', \epsilon - \delta) \quad\quad\quad\quad\quad\quad\quad\quad\quad \text{by Equation 1}
\end{aligned}
$$

This is sufficient to conclude that we have `cert`$(x', \; \epsilon - \delta)$. $\qquad\square$

GloRo Nets are Lipschitz-based, therefore they are monotonic by Lemma A.5. Aside from the fact that nondeterminism/uncertainty arises because of the need to sample to evaluate the smoothed function, Randomized Smoothing provides the exact robustness radius on the smoothed function. That is, in the infinite sample limit, Randomized Smoothing is an exact method, and therefore monotonic by Lemma A.3; though in practice this property only holds with high probability (i.e., the probability can be bounded from below). We note, however, that the robustness guarantees provided by Randomized Smoothing are analogously probabilistic.

## B  Lower bounds on attack efficacy (more results)

### B.1  Efficacy of a modified version of Algorithm 3.1

We evaluated the degradation attack efficacy of a modified version of Algorithm 3.1. This attack (Algorithm B.1) also invokes an off-the-shelf white-box attack algorithm (PGD), indicated by `Attack`, but to find an adversarial example $x''$ in a ball of radius $2\epsilon$ centered at $x$ (line 2). Next, $x''$ is simply projected onto the $\epsilon$-sphere centered at $x$ (line 3). For $\ell_2$, `Project` is defined as $x + \min\{\epsilon \;,\; ||x'' - x||_p\} \cdot \frac{x'' - x}{||x'' - x||_p}$, whereas for $\ell_\infty$ we simply need to clip $x''$.

Table 3 presents the results of using this modified algorithm for degradation attacks against GloRo Net models. The models, datasets, and $\epsilon$ values are the same as used for evaluating the efficacy of Algorithm 3.1. The efficacy of this attack is similar to Algorithm 3.1 (presented in Table 1).

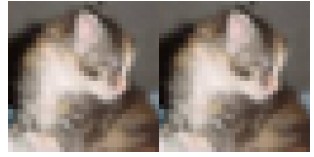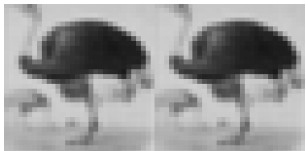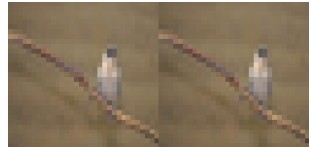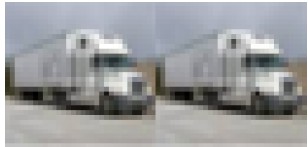

Figure 4: Visualizations of successful degradation attacks on CIFAR-10 ($\epsilon = 0.141$).

Table 4: Lower bounds for false positive rates induced by degradation attack against KW.

|  | baseline VRA (%) | false positive rate (%) | utility reduction (%) |
|---|---|---|---|
| *MNIST ($\epsilon = 0.1$)* | 95.6 | 95.2 | 91.0 |
| *CIFAR-10 ($\epsilon = 2/255$)* | 52.5 | 16.8 | 8.8 |

Table 5: Upper bounds for false positive rates induced by degradation attack against KW.

|  | baseline VRA (%) | false positive rate (%) | utility reduction (%) |
|---|---|---|---|
| *MNIST ($\epsilon = 0.1$)* | 95.6 | 100.0 | 95.6 |
| *CIFAR-10 ($\epsilon = 2/255$)* | 52.5 | 64.8 | 34.0 |

### B.2 Visualization of successful degradation attack inputs

Figure 4 provides samples of inputs that constitute successful degradation attacks. In each pair of images, the original image is shown on the left and the perturbed image (the attack) is shown on the right. As expected, the pairs of images are visually indistinguishable; thus the adversary can be considered to have successfully delivered an *imperceptible* attack.

## C Degradation attacks against $\ell_\infty$ defenses

In this section, we evaluate the efficacy of degradation attacks against models defended with a $\ell_\infty$ certified defense. In particular, we consider models trained and defended using the convex outer adversarial polytope approach of Wong & Kolter (2018), that we refer to as KW. We compute lower and upper bounds in a similar manner as described in Section 4, using KW for the $(\epsilon, \ell_\infty)$-local robustness check. For the lower bounds, the set $\underline{test_A}$ is constructed using Algorithm 3.1 where PGD is used for `Attack`. We use the pre-trained models made available by the authors of KW.[5] The $\epsilon$ values are the same as used by Wong & Kolter (2018).

Table 4 reports the lower bounds, i.e., the efficacy of Algorithm 3.1 against KW. Table 5 reports the upper bounds. We see that $\ell_\infty$ defenses like KW can be extremely susceptible to degradation attacks—on MNIST, the simple attack given by Algorithm 3.1 succeeds approximately 95% of the time, and the upper bounds indicate that it may be possible for a more sophisticated adversary to succeed *100% of the time*. On CIFAR-10, the attack success rates are more modest, partially because of the small value of $\epsilon$.

While striking, particularly for MNIST, these results are not entirely unexpected, given the geometry of $\ell_\infty$ space. Defenses based on robustness certification are susceptible to degradation attacks unless the underlying model is $(2\epsilon, \ell_p)$-robust. In $\ell_\infty$ space, the volume contained by a ball overlaps much more closely with $[0,1]^d$ (the typical the domain for, e.g., image data), which intuitively means that the domain is "used up" much more quickly as the required robustness radius increases. By contrast, in high-dimensional Euclidean space, the radius can be as large as $\sqrt{d}$ before it necessarily envelops the entire domain.

---

[5]http://github.com/locuslab/convex_adversarial

# D  Statement of broader impact

Our work sheds light on existing vulnerabilities in state-of-the-art certifiably robust neural classifiers. These degradation attacks could be deployed by malicious entities to degrade the utility of deployed models. However, by putting this knowledge out in the public domain and making practitioners aware of the existence of degradation attacks, we hope that precautions can be taken to protect existing systems. Moreover, it highlights the need to harden future systems against such attacks.

