# OpenReview forum: "Degradation Attacks on Certifiably Robust Neural Networks"
_TMLR — Accepted by TMLR_

### Review · Reviewer_8nCi · 2022-07-27

**Summary Of Contributions:**

The paper studies certifiably robust classifiers---i.e., classifiers that for a given inference-time input, attempt to verify if all small perturbations of that input are classified consistently and potentially abstain if not.

The authors argue that such classifiers suffer from a drawback: if a small perturbation of the given input is classified differently, but that perturbation is not part of the space of valid inputs (i.e., “the data manifold”), then the classifier might abstain unnecessarily. The authors argue that this reduces the utility of the classifier as it requires a human operator to intervene.

In order to support this argument, the authors consider existing certifiably robust classifiers and develop “degradation attacks”: perturbations that cause the classifier to abstain (since the classifier is not robust around the perturbed input), even though the classifier is correct and robust on the original input. In terms of methodology, these attacks rely on existing methods for crafting adversarial examples (e.g., PGD). They show that for standard datasets an adversary can cause the classifier to abstain on a significant fraction (e.g., ~30%) of test inputs that are normally correctly classified and certifiably robust.

**Broader Impact Concerns:**

The paper discusses methods for degrading the performance of ML systems. Thus there is the potential that such methods could be used to cause harm. At the same time, in its current form, the attack proposed does not seem to pose a perceivable threat.

**Requested Changes:**

My concerns are not specific to any particular part of the paper but rather to the broader motivation and approach.

**Strengths And Weaknesses:**

The authors propose an attack that is, to the best of my knowledge, novel. The paper is well-written and the methodology sound.

At the same time, the threat model considered by the paper is not well motivated.

On one hand, the argument that classifiers abstain unnecessarily because they ignore the “data manifold” is non-actionable. For a certifiable method to take advantage of the domain of valid images, one would need access to a mathematically precise definition of what constitutes a valid image. Note that even methods that aim to approximate the “data manifold” cannot be used to actually certify that certain inputs are on the manifold in any meaningful sense.

On the other hand, the significance of the proposed degradation attacks is questionable. After all, the certifiable robust models are working exactly as intended: they reject inputs on which the base model’s prediction is not robust. From that perspective, I find a significant part of the paper narrative misleading. For instance, terminology such as “unnecessary rejections” is misleading: if these inputs are not robust then why should they be accepted by a robustness certification method?

Moreover, from a deployment point of view, it is reasonable to expect the overall system to behave as follows: the system is predicting accurately on natural inputs and abstains on ones that have been adversarially perturbed. The proposed degradation attacks do not change this behavior at all as the model still rejects corrupted inputs. At an even higher level, it is seems hard to imagine settings where the model designer is ok abstaining on certain suspicious inputs and at the same time the adversary can cause a significant utility degradation by causing the model to abstain. It would be useful for the discussion if the authors could provide examples of real-world threats that are at least somewhat plausible.

Overall, I do not find the arguments presented by the paper particularly convincing and thus I don’t believe that it moves the discussion around adversarial robustness forward.

---

> ### Author Response · Authors · 2022-07-27
> **Response to Reviewer 8nCi**
>
> We thank the reviewer for their detailed comments.
>
> We disagree with the comment that “certifiable robust models are working exactly as intended”. If the goal is to certify robustness, then the comment is accurate but our point in the paper is precisely that “defending against adversarial examples” is not equivalent to “certifying local robustness”.
>
> Consider the oft-cited example of an image classifier deployed in an autonomous vehicle and tasked with detecting STOP signs on the road. Given a “natural” image with a STOP sign that can be handled correctly by our classifier, when the adversarial ML community says that the classifier should be free from adversarial examples, it means that for every “perturbed” or “corrupted” version of the natural image, the classifier should continue to predict correctly. Notice here that we do not want the system to simply abstain on all “corrupted” or perturbed inputs. Instead, we want the system to continue predicting on perturbed inputs as long as the perturbations do not lead to a change in the prediction.
>
> Degradation attacks shed light on the scenario where a perturbation does not cause any change in the prediction of the classifier yet the perturbed or corrupted input gets rejected. Describing such rejections as “unnecessary” is justified in light of the fact that the classifier should continue predicting as long as the perturbations or corruptions are harmless.
>
> Again consider the example of the classifier in the autonomous vehicle. Assume that whenever the classifier abstains from making a prediction, the vehicle has to come to a standstill (to ensure safety). Let us say that our classifier is highly robust, i.e., in most cases, perturbations or corruptions of natural images do not cause it to change its prediction. The system designer knows this but, for guaranteed protection, they deploy the classifier along with a run-time robustness check. This ensures safe handling of the relatively rare scenarios where a perturbation of a natural image causes the classifier to change its prediction.
>
> Our degradation attacks show that even for a highly robust classifier, it is easy to craft perturbed inputs where the classifier prediction does not change, but a robustness check fails. While the system designer expected the fail-safe behavior of the vehicle coming to a standstill to only be rarely invoked, degradation attacks can cause the fail-safe behavior to be triggered much more frequently. Thus, this is a scenario where the system designer was okay with the model abstaining (since they expected this to be a rare event), but degradation attacks turned this into a frequent event.

---

> > ### Comment · Reviewer_8nCi · 2022-07-30
> > **Response**
> >
> > I thank the reviewers for their response and provided justification.
> >
> > > “defending against adversarial examples” is not equivalent to “certifying local robustness”
> >
> > Precisely. A model designer choosing to deploy a certifiably robust defense with abstention, does not just care about defending against adversarial examples, but being confident that all predictions are stable. Since in the case of degradation attacks the model predictions _are not stable_ (there exist points within the $\epsilon$-ball of the perturbed example that are classified incorrectly) the certified classifier rejects them. Thus the classifier is behaving correctly under the higher deployment standards.
> >
> > > whenever the classifier abstains from making a prediction, the vehicle has to come to a standstill (to ensure safety)
> >
> > That's an interesting point, perhaps worth incorporating into the manuscript. Nevertheless, I still find it rather contrived, since I find it hard to imagine a scenario where such behavior is actually implemented in the real world.

---

> > > ### Author Response · Authors · 2022-07-31
> > > **Response to Reviewer 8nCi**
> > >
> > > We thank the reviewer for engaging in this interesting discussion.
> > >
> > > > A model designer choosing to deploy a certifiably robust defense with abstention, does not just care about defending against adversarial examples, but being confident that all predictions are stable
> > >
> > > Our reasoning is as follows - Researchers noticed that neural networks are vulnerable to adversarial perturbations. To defend against adversarial attacks, researchers have proposed defense methods based on certifying local robustness. Thus it is not the case that researchers deploy certifiable robust defenses because they want **all** predictions to be stable but rather because that is the **only known way** to defend against adversarial attacks in a certifiable way. However we show that this kind of defense is too strong. Through our work we point out that there is a need for new research that goes beyond local robustness as a means of certifiably defending against adversarial attacks.
> > >
> > > > That's an interesting point, perhaps worth incorporating into the manuscript. Nevertheless, I still find it rather contrived, since I find it hard to imagine a scenario where such behavior is actually implemented in the real world.
> > >
> > > We will incorporate this point more clearly in the paper. As for the example being contrived, we respectfully disagree. The notion of an autonomous vehicle coming to a standstill in case of unsafe situations has been recommended in recent regulations [1]. In particular, for Level 3 autonomous vehicles with Automatic Lane Keeping Systems (ALKS), the ALKS can issue transition demands requesting the driver to take over the driving task when the car approaches traffic conditions outside its Operating Design Domain. If the driver is unresponsive or becomes inattentive and ALKS-issued alerts meant to restore driver attentiveness are ineffective, a minimum-risk manoeuvre (e.g., **bringing the car to a standstill**) will be performed.
> > >
> > > But also consider another example. Let’s say that a neural classifier is deployed as a fake news detector. In this case, if a certifiably robust defense is deployed at run-time, then an adversary can craft degradation attacks that cause the fake news detector to hand-over decision making to a human operator, even for articles that the classifier correctly identifies as fake news. Given the volume of articles that are posted on social media sites, such a system can only function if the human operators are invoked in extremely rare cases. By forcing the fake news detector to frequently abstain, an adversary can basically overwhelm the human operators and render the system useless.
> > >
> > > [1] UNECE. ECE/TRANS/WP.29/2020/81—United Nations Regulation on Uniform provisions concerning the approval of vehicles with regard to Automated Lane Keeping Systems, June 2020.

---

> > > > ### Comment · Reviewer_8nCi · 2022-07-31
> > > > **Response**
> > > >
> > > > > certifiable robust defenses [...] is the only known way to defend against adversarial attacks in a certifiable way.
> > > >
> > > > I find this statement to be somewhat misleading. Certifiable defenses can be used to either:
> > > > - Certify that a model is robust on a fraction of the test set. A model that is deemed certifiably robust in this way can be trusted to be robust to adversarial examples on a fraction of test inputs (as long as they follow the test distribution), in a certifiable manner. Such a model does not use any certification during deployment and thus the degradation attacks proposed would not affect it. Nevertheless, this model is defending against adversarial attacks in a certifiable way.
> > > > - Certify that an individual instance is robust to perturbations by attempting to certify during inference. This is a significantly stricter robustness requirement. The guarantee conveyed is that: (a) either an input is robust and thus no adversarial versions exist, (b) there exist small perturbations of that input that induce a different prediction. A model designer can choose to use that method to reject inputs in the case of (b), out of an abundance of caution, in order to ensure that an adversary can at most cause the model to abstain. However, in that case, the designer explicitly chooses to abstain on inputs where the predictions are not stable, and thus the fact that the model abstains on degradation attacks (which produce non-stable inputs) should not come as a surprise---it is an explicit design choice.
> > > >
> > > > > As for the example being contrived, we respectfully disagree.
> > > >
> > > > I apologize if my point did not come across clearly. My concern here is that there are so many ways in which deployment conditions can deviate from what is expected, that I find it hard to believe that someone would rely on an adversarial examples certification method to ensure the smooth operation of an AV.
> > > >
> > > > In fact, there are so many ways in which nature or an adversary could create conditions where the ML systems would produce a _stable_, yet incorrect prediction, in which case certification methods are ineffective. Thus I find arguing about robustness to $\epsilon$ vs. $2\epsilon$ $\ell_2$-balls in an AV setting to be of limited relevance.
> > > >
> > > > Similarly, in the setting of fake news detection, if the goal is to produce articles that are ambiguous to the classifier and thus need human intervention, why would one rely on small perturbations of natural inputs, instead of directly constructing a different attack?
> > > >
> > > > In any case, at this point, this discussion around hypothetical scenarios might not be productive. Since our disagreement is not factual, I'd rather defer to the AE to judge the relevance of the setting.

---

> > > > > ### Author Response · Authors · 2022-08-01
> > > > > **Response to reviewer**
> > > > >
> > > > > > A model that is deemed certifiably robust in this way can be trusted to be robust to adversarial examples on a fraction of test inputs (as long as they follow the test distribution), in a certifiable manner. [...] Nevertheless, this model is defending against adversarial attacks in a certifiable way.
> > > > >
> > > > > In this case, at best, the model provides a statistical guarantee of being defensible against adversarial attacks. Declaring that such a model is defending against adversarial attacks in a *certifiable way* is overly strong since there is no actual robustness proof for *all* the in-distribution inputs.
> > > > >
> > > > > > However, in that case, the designer explicitly chooses to abstain on inputs where the predictions are not stable.
> > > > >
> > > > > Are model designers who deploy local robustness checks during inference doing so as a defense against adversarial examples or to ensure that model predictions are stable? If it’s the latter, then we agree with the reviewer that the model is behaving as designed on degradation attack inputs. However, if it’s the former, then we maintain that a defense based on local robustness is too strong - the model predictions only need to be stable with in-distribution inputs that are $\epsilon$-close and not with all $\epsilon$-close inputs.
> > > > >
> > > > >
> > > > > But we agree with the reviewer that our disagreements are perhaps no longer factual and we have reached an impasse.

---

### Review · Reviewer_Br6u · 2022-07-27

**Summary Of Contributions:**

This paper investigates the over-cautiousness of existing certifiable defenses and proposes corresponding degradation attacks to cause frequent rejections in run time. Empirical experiments show good results of the proposed attacks on two types of certified defenses GloRo nets and randomized smoothing.

**Broader Impact Concerns:**

Since the paper proposes new attacks aiming to break existing certifiable defenses, it would be beneficial to add a Broader Impact Statement.

**Requested Changes:**

Changes:
1. The proposed SPGD is similar to [A]. A discussion could be helpful.
2. To demonstrate the usefulness and generalization of the proposed attack, the proposed attack should be discussed and evaluated with other SOTA complete verifiers like alpha-beta-CROWN [B, C] (the winner of the latest neural network verification competition) or incomplete ones like K&W [D], and certifiable training methods like CROWN-IBP [E].

[A] Provably Robust Deep Learning via Adversarially Trained Smoothed Classifiers, NeurIPS 2019

[B] Beta-CROWN: Efficient Bound Propagation with Per-neuron Split Constraints for Complete and Incomplete Neural Network Verification, NeurIPS 2021

[C] Fast and Complete: Enabling Complete Neural Network Verification with Rapid and Massively Parallel Incomplete Verifiers, ICLR 2021

[D] Provable defenses against adversarial examples via the convex outer adversarial polytope, ICML 2018

[E] Towards Stable and Efficient Training of Verifiably Robust Neural Networks, ICLR 2020

**Strengths And Weaknesses:**

Strength
1. This paper points out a very interesting problem of certified defense especially with GloRo nets and Randomized smoothing when applied in run-time applications. The empirical results well support the claim. This encourages a more general (certifiable) robustness definition and more study on global robustness.
2. Solid theoretical attack efficacy upper and lower bounds are analyzed.
3. The paper is well presented and easy to follow.
4. Two potential defenses are provided and discussed.

Weakness
1. The proposed methods themselves have very limited novelty though the angle for applying these attacks is new.
2. In practice, verification is a plus for the model robustness evaluation. It is safe to trust the results if a sample is verified but a sample that cannot be verified does not necessarily mean we have to reject it. There are many other ways to measure the robustness and give confidence scores of the predictions. Also, over-cautiousness is the intrinsic property of certifiable defenses and should be the correct thing to have. Therefore, the degradation attack is useful but not such a severe problem as described in the paper.
3. Limited experimental evaluation (see requested changes below)

---

> ### Author Response · Authors · 2022-08-14
> **Response**
>
> We thank the reviewer for their comments.
>
> 1. We will include a discussion of the relationship between our SPGD attack and [A].
>
> 2. We already evaluate our attack against the K&W defense. These results are presented in Appendix C. GloRo Nets, Randomized Smoothing, and K&W represent three state-of-the-art certification methods that use very different certification algorithms. The fact that our attack is effective against all these certified defenses demonstrates its general applicability.

---

> > ### Comment · Reviewer_Br6u · 2022-08-17
> > **Response**
> >
> > Thank the authors for the response, which mostly addressed my questions. Additional evaluation with SOTA complete verifiers would be a plus to make the results more convincing.

---

### Review · Reviewer_SUEh · 2022-08-13

**Summary Of Contributions:**

The authors find that certified defenses against adversarial attacks may be overly cautious in that they reject examples which are not adversarial. In particular, they define a set $M$ (of valid inputs) and show that certified defenses might reject an example which is not locally robust but is consistent with all $\epsilon$-close inputs in $M$. They then develop a "degradation" attack which causes certified defenses to reject examples that could have been correctly classified. They apply their experiments on the MNIST, CIFAR-10, and ImageNet datasets.

**Requested Changes:**

I think to improve this paper, you need a thorough section explaining
1. What is $M$, and how does the definition of $M$ relate to current notions of adversarial robustness?
2. What does the adversary know about $M$?
3. Why is your attack different than just finding an adversarial example that is $2\epsilon$ away?
4. How is $M$ defined for your experiments?

In terms of writing, the paper is a bit repetitive. Section 2, Section 3.1, and Section 3.3 all explain the degradation attack. I would condense these sections, and explain the attack clearly (once).

**Strengths And Weaknesses:**

## Strengths
The problem is certainly interesting. A main failing of certified defenses are that they are too conservative. I think that fully characterizing what type of inputs certified defenses are too cautious towards is a worthwhile endeavor.

## Weaknesses
I don't think this paper is well-posed. Most of my concerns have to do with the set $M$, which is central to both their overall claim (that certified defenses are overly cautious) and their degradation attack.

**Defining M**: $M$ is not well-defined. What does it mean to be a valid input? Do adversarial examples count as valid inputs? Depending on how you define $M$, these certified defenses are either overly cautious or appropriately rejecting risky examples.

**Assumptions of the adversary**: Even if we could define $M$, it is unclear to me why the adversary would know $M$ but the certified defense would not. These assumptions are not justified in the paper.

**Assumption in Section 3.3** Why can't $x'$ be an adversarial input? I realize that most image datasets are well-separated, but (again) this statement makes a strong assumption about what $M$ is. Indeed, it seems like you are making the assumption that anything within $2\epsilon$ of a data point that has a different class label is not a valid input. If you make this assumption, robustness is already solved!

**The attack**: At the end of the day, the degradation attack seems like it is just performing an adversarial attack that is $2\epsilon$ away instead of $\epsilon$ away, when up against a defense with $\epsilon$ robustness. That this works is not that interesting. Also, what are you using as $M$ in your empirical experiments?

**Significance** The fact that certified defenses are overly cautious is already pretty well known (indeed, certified accuracy is almost always much lower than clean accuracy for almost every defense). It would have been interesting to characterize this more fully, but I don't think this paper puts forward an actionable framework.

---

> ### Author Response · Authors · 2022-08-14
> **Response (Part 1)**
>
> We thank the reviewer for their comments.
>
> ### Defining M:
> > What is $M$, and how does the definition of $M$ relate to current notions of adversarial robustness? What does it mean to be a valid input? $M$ is not well-defined.
>
> The notion of *valid* inputs was not invented by us but comes from the literature on adversarial perturbations [Szegedy et al., 2014; Carlini & Wagner, 2017]. We reproduce the definitions from [Szegedy et al., 2014; Carlini & Wagner, 2017] in our paper (see Definition 2.1) which define adversarial perturbations against *valid* inputs. We merely made the definition more precise, by defining $M$ as the set of valid inputs.
>
> More precisely, let $D$ be the input distribution for the classification problem, i.e., if the goal of the classification problem is to learn a function from $X$ to $Y$ given $(x,y)$ samples, then $D$ is the marginal distribution over $X$ from which the inputs are drawn. $M$ is the support set of $D$.
>
> An input $x$ is valid if $x \in M$, i.e., $x$ is in-distribution. An input $x’$ is an adversarial example if there exists an $\epsilon$-close *valid* input $x$ such that $f(x) \neq f(x’)$ where $f$ is the classifier. Notice that this definition does not specify whether $x’$ is a valid input or not. This definition of an adversarial example is consistent with existing definitions in the literature but we make it explicit that $x$ needs to be a valid input via set $M$.
>
> [Szegedy et al., 2014] Christian Szegedy, Wojciech Zaremba, Ilya Sutskever, Joan Bruna, Dumitru Erhan, Ian J. Goodfellow, and Rob Fergus. Intriguing properties of neural networks. In Yoshua Bengio and Yann LeCun (eds.), 2nd International Conference on Learning Representations, ICLR 2014, Banff, AB, Canada, April 14-16, 2014, Conference Track Proceedings, 2014
>
> [Carlini & Wagner, 2017] ​​N. Carlini and D. Wagner. Towards evaluating the robustness of neural networks. In 2017 IEEE Symposium on Security and Privacy (SP), pp. 39–57, Los Alamitos, CA, USA, may 2017. IEEE Computer Society. doi: 10.1109/SP.2017.49.
>
> ### Assumptions of the adversary:
> > What does the adversary know about $M$? It is unclear to me why the adversary would know $M$ but the certified defense would not.
>
> It is not true that the adversary knows $M$ and the defender does not. Both the adversary and the defender can only draw samples from $M$.
>
> The only assumption in our work is that the adversary can see unperturbed inputs $x$ generated by Nature (e.g. images taken with a camera) and these unperturbed inputs are in-distribution  samples drawn from the input distribution $D$, i.e., $x \in M$. These are very standard assumptions.
>
> Given an input $x$, an adversary that uses our attack algorithms simply tries to find an input $x’$ such that $|| x - x’|| \leq \epsilon$ and the model cannot be certified $\epsilon$-locally robust at $x’$.
>
>
> ### Assumption in Section 3.3:
> > If you make this assumption, robustness is already solved!
>
> This is not correct. It is true that, in our evaluation, we assume that the datasets are well-separated such that any two valid inputs with a different ground-truth label are separated by at least $2\epsilon$. This is an empirically well-justified [Yang et al., 2020b] assumption about the nature of $M$, as the reviewer already notes. However, this assumption does not imply that robustness is solved since it says nothing about the classifier that is learned from data. It only implies that there exists a classifier with 100% verified robust accuracy (VRA). Algorithmically finding such a classifier is another matter, and state-of-the-art certifiably robust classifiers exhibit a much lower VRA even on image datasets where the separability assumption holds.
>
> Perhaps this confusion is caused due to the definition of $test_R$. $test_R$ is the set of test inputs where the model is certified to be $\epsilon$-locally robust *and* also known to be accurate. We will clarify this in the text.
>
> [Yang et al., 2020b] Yao-Yuan Yang, Cyrus Rashtchian, Hongyang Zhang, Russ R Salakhutdinov, and Kamalika Chaudhuri. A closer look at accuracy vs. robustness. In NeurIPS, 2020

---

> > ### Author Response · Authors · 2022-08-14
> > **Response (Part 2)**
> >
> > ### The attack:
> > > How is $M$ defined for your experiments?
> >
> > We do not explicitly define the full $M$. We only assume that the inputs in the test dataset are drawn from  $M$. Assuming that the train and test datasets are drawn from the same distribution is a standard assumption of machine learning. We also assume that any two inputs in $M$ with a different ground-truth label are separated by at least $2\epsilon$.
> >
> > > Why is your attack different than just finding an adversarial example that is $2\epsilon$ away?
> >
> > We agree that the operational details of our attack algorithm are similar to an algorithm for finding an adversarial example in the $2\epsilon$ ball (Technically, our attack is not just finding an adversarial example in the $2\epsilon$ ball. Instead, it tries to find an input $x’$ in the $\epsilon$ ball such that the model cannot be certified $\epsilon$-locally robust at $x’$).
> >
> > However, this similarity actually reinforces the key point that we are trying to convey. The use of a run-time local robustness check as a defense against adversarial examples allows an $\epsilon$-bounded adversary to exploit adversarial examples in a $2\epsilon$ ball. Put differently, even if the model is free from $\epsilon$ adversarial examples (i.e., the model is $\epsilon$-locally robust at all points in $M$, as per Theorem 2.4),  the presence of a run-time $\epsilon$-local robustness check allows an $\epsilon$-bounded adversary to exploit model susceptibility to $2\epsilon$ adversarial examples. To be protected from such degradation attacks, the model is then forced to exhibit $2\epsilon$-local robustness at all points in $M$.
> >
> >
> > ### Significance:
> > > The fact that certified defenses are overly cautious is already pretty well known (indeed, certified accuracy is almost always much lower than clean accuracy for almost every defense).
> >
> > It is well-known that certified defenses are overly cautious in the sense that local robustness certifiers are typically sound but not complete, i.e., while a certificate of local robustness implies that the model is indeed locally robust, the failure to find such a certificate does not imply that the model is not locally robust.
> >
> > However, our  contribution is to show that even exact (i.e., sound and complete) local robustness certifiers can be overly cautious if deployed at run-time since the model predictions only need to be consistent with valid (i.e., in-distribution) inputs that are $\epsilon$-close and not with all $\epsilon$-close inputs.
> >
> > As an aside, the fact that certified accuracy is almost always much lower than clean accuracy for almost every defense is not necessarily evidence for the cautiousness of certified defenses. A model (with high clean accuracy) can exhibit lower certified accuracy simply because of the fact that the model  is actually susceptible to adversarial examples.

---

> > > ### Comment · Reviewer_SUEh · 2022-08-16
> > > **Further Questions**
> > >
> > > Hi,
> > >
> > > Thanks for your response.
> > >
> > > > On the definition of $M$
> > >
> > > We do not have a precise definition of what the input distribution $D$ is for images. The defender does not know what the data manifold is: thus, they need to assume that any potential input is a feasible image. Since the defender does not know what $M$ is (and indeed, changing $M$ would change your definition of robustness), I don't see the claims as actionable. In this, I agree with Reviewer 8nCi, who makes a similar point.
> > >
> > > > On the assumptions of the adversary
> > >
> > > Whether the "attack" is a degradation attack (versus the certified defense correctly rejecting an adversarial example) depends on the definition of $M$. Here, it seems that your degradation attack is based on the assumption that every ground truth input is at least 2$\epsilon$ apart. It seems like you need this assumption (otherwise the generated $x'$ from the degradation attack would be an actual adversarial example instead of a degradation attack). But the certified defense is not using this assumption, which is why it seems like the adversary knows more about $M$ than the defense.
> > >
> > > > Similarity of the attack to finding an adversarial example that is 2$\epsilon$ away
> > >
> > > Correct me if I'm wrong: your attack involves finding an example $x'$ in the $\epsilon$ ball of $x$ which cannot be certified locally $\epsilon$ robust. At its core, means that you found an example $x''$ which was between $\epsilon$ and $2\epsilon$ away from $x$ but does not share the same label as $x$. This seems equivalent to finding an adversarial example (when $M$ is all possible images) in the $2\epsilon$ ball for examples $x$ that are robust with $\epsilon$.

---

> > > > ### Author Response · Authors · 2022-08-17
> > > > **Response**
> > > >
> > > > > The defender does not know what the data manifold is: thus, they need to assume that any potential input is a feasible image.
> > > >
> > > > If one assumes that *any* potential input is a feasible image then the definition of robustness becomes $\forall x, x' \in R^d~s.t. ||x-x'|| < \epsilon \Rightarrow f(x)=f(x')$. This can be satisfied only by making $f$ the constant function, rendering the classifier $f$ useless. Hence if the goal is certification of robustness for a classifier model, one needs to consider the set of valid inputs.
> > > >
> > > > > I don't see the claims as actionable.
> > > >
> > > > Our observations lead to the following concrete actionable interventions:
> > > >
> > > > 1.  Model developers that use certified run-time defenses based on local robustness should evaluate their models with $2\epsilon$ robustness even when the adversary is $\epsilon$-bounded. Verified robust accuracy (VRA) numbers calculated wrt $\epsilon$ robustness are overly optimistic and do not reflect the performance (% of inputs where the model is both accurate and does not reject) one would see for the deployed model.
> > > >
> > > > 2. If it turns out that the VRA is too low when evaluated at $2\epsilon$, the developer can configure the certified defense to enforce local robustness at $\epsilon’$ where $\epsilon’ < \epsilon$. This reduces the vulnerability to degradation attacks at the cost of an increased vulnerability to adversarial attacks.
> > > >
> > > > 3. As suggested in Section 4.1, the developers can train the model to be robust wrt $2\epsilon$ perturbations when faced with an $\epsilon$-bounded adversary in order to defend against degradation attacks.
> > > >
> > > >
> > > > > It seems that your degradation attack is based on the assumption that every ground truth input is at least $2\epsilon$ apart.
> > > >
> > > > Please note that degradation attacks can be successful even when the data is not generally well-separated. We only use this assumption in our experiments for simplifying the calculations and measurements. The adversary does not know or make any well-separateness assumptions (and neither do our attack algorithms).
> > > >
> > > > Note that in our experiments, we do not assume that every ground truth input is at least $2\epsilon$ apart but rather any two inputs with *different* ground truth labels are at least $2\epsilon$ apart (i.e., inputs with the same labels can be less than $2\epsilon$ apart). Further for test inputs $x \in test_A$ (i.e., $x$ is a test input such that the model is $\epsilon$-locally robust and accurate at $x$, and there exists an input $x’$ such that $||x-x’|| \leq \epsilon$ and the model is not $\epsilon$-locally robust at $x’$), the model is accurate on all valid inputs (if any) in the $2\epsilon$ ball centered at $x$. We will clarify this in Section 3.3.
> > > >
> > > >
> > > > > Similarity of the attack to finding an adversarial example that is $2\epsilon$ away
> > > >
> > > > Yes, our attack is similar to a $2\epsilon$ attack but it is a surprising and new observation that $2\epsilon$ attacks are even relevant for an $\epsilon$-bounded adversary.

---

> > > > > ### Comment · Reviewer_SUEh · 2022-08-17
> > > > > **Some clarifications**
> > > > >
> > > > > Hi,
> > > > >
> > > > > Thank you for your response:
> > > > >
> > > > > ### Assuming any potential input is a feasible image
> > > > >
> > > > > While a classifier cannot be robust on all points (as indeed this would need to be a constant classifier), it doesn't make sense for a certified defender to accept a point for which the model is not locally robust. Distinguishing between a point which cannot be certified as robust and an adversarial example requires having knowledge of $M$ (depending on how you define $M$, your non-robust point becomes an adversarial example). Thus, since the defender cannot assume what $M$ is, it needs to treat any potential image as a feasible input and thus reject points that cannot be certified as locally robust. In other words, rejecting points that are not stable seems like very reasonable behavior.
> > > > >
> > > > > ### Adversary's knowledge
> > > > >
> > > > > To clarify, I meant that the adversary assumes that every input with different groundtruth labels is $2\epsilon$ apart. The reason that this seems to be the case is this line on page 6
> > > > >
> > > > > > "Since the model is certified locally robust at x, we know that $x$ and $x'$ share the same label and $x'$ is not an
> > > > > adversarial input (under the empirically justified assumption that there exists no other point of a different class label in $M$ that is $\epsilon$ close to $x$ (Yang et al., 2020b))."
> > > > >
> > > > > Thus, the only reason that $x'$ is a degradation attack is because $x'$ is a point for which the model is not locally robust but is not an adversarial example (according to this empirical assumption on $M$). If you didn't have this assumption, $x'$ could very well be an adversarial example (and the defender would be correct in rejecting it).
> > > > >
> > > > > Thus, isn't the adversary making an assumption on $M$? Please clarify.

---

> > > > > > ### Author Response · Authors · 2022-08-17
> > > > > > **Response**
> > > > > >
> > > > > > Thank you for engaging in this discussion.
> > > > > >
> > > > > > ### Assuming any potential input is a feasible image
> > > > > >
> > > > > > While indeed current certifiers (i.e., run-time local robustness checks) operate without using the knowledge of $M$ (and this may be reasonable) our results indicate that for an $\epsilon$-robust model (and by that we mean a model that is $\epsilon$-locally robust on *all valid* inputs), current certifiers will be unable to certify it. They will only be able to certify $\epsilon$-robustness for models that are actually $2\epsilon$-robust.
> > > > > >
> > > > > > Even without the knowledge of $M$, model developers can (and should) take some concrete actionable steps to mitigate the threat of degradation attacks (as described in our previous response) while continuing to use run-time robustness checks.
> > > > > >
> > > > > > ### Adversary's knowledge
> > > > > >
> > > > > > Let’s look at the situation from the adversary’s perspective. The adversary only assumes that they get an input $x \in M$, and given such an input, the adversary constructs an input $x’$ such that (i) $|| x-x’ || \leq \epsilon$, (ii) the model is not $\epsilon$-locally robust at $x’$. At this point, the adversary does not know whether $x’$ is a successful degradation attack or not because they are making no assumptions about $M$. The adversary is only hoping that the crafted input will lead to a degradation attack. It is only during empirical evaluation, where we actually need to know whether the attack was successful or not, that we have to make assumptions about $M$.
> > > > > >
> > > > > > Moreover, for the sake of empirical evaluation, one could make other assumptions about $M$ and the model $f$. For instance, given original input $x$ where $f$ is $\epsilon$-locally robust and an adversarially-generated input $x’$, one could assume that for all *valid* $x’’$ in the $\epsilon$ ball centered at $x’$, $f(x’’)=f(x)$. Under such an assumption, $x’$ would be a successful degradation attack input. In this case, the assumption is only about $f$ and not about $M$.
> > > > > >
> > > > > > The particular assumption we make about $M$ is motivated by its simplicity and empirical justifiability.

---

### Decision · Action_Editors · 2022-09-19

**Recommendation:** Accept with minor revision

**Comment:**

I would like to thank the authors for their submission to TMLR. I hope the following feedback, together with the reviews and discussion, will provide some insights in the outcome and a way forward for the paper.

This manuscript builds on the notion of valid inputs [Szegedy et al., 2014; Carlini & Wagner, 2017] to define the set of valid inputs $M$. It then also makes the assumption that any two inputs in $M$ with a different ground-truth label are separated by at least $2\varepsilon$. The paper observes that when a model exhibits $\varepsilon$ robustness, it is susceptible to a degradation (i.e., availability) attack if the model is not $2\varepsilon$ robust. Put altogether, this implies that “even exact (i.e., sound and complete) local robustness certifiers can be overly cautious if deployed at run-time since the model predictions only need to be consistent with valid (i.e., in-distribution) inputs that are $\varepsilon$-close and not with all $\varepsilon$-close inputs” as summarized by the authors.

## Criteria

I assessed the submission and reviews based on the two criteria listed by TMLR.

> Are the claims made in the submission supported by accurate, convincing and clear evidence?

Reviewers did not find any issues with the accurateness of the submission's claims. That said, all reviewers ended up disagreeing with the authors about the significance of their claims and the practicality of the threat model considered in the manuscript. I would therefore request that the authors clearly state the limitations of their work in the abstract, introduction, section 4.1, and conclusion. This would include:

* A detailed discussion and acknowledgment describing how the lack of knowledge of $M$ makes the findings of this manuscript less actionable.
  * For instance, the authors write “the developers can train the model to be robust wrt $2\varepsilon$ perturbations when faced with an $\varepsilon$-bounded adversary in order to defend against degradation attacks”: what would be the impact on accuracy?
  * Alternatively, how much lower the value of $\varepsilon’$ used for certification would have to be in order to avoid degradation attacks?
  * There are numerous excellent points that were made during the discussion between authors and reviewers, I would ask that the authors carefully go through these points and integrate them in the revised manuscript.

* A clearly marked paragraph/subsection containing a threat model. I would pay particular attention to the discussion with Reviewer SUEh when writing the threat model.

* A discussion of the dichotomy between “defending against adversarial examples” and “certifying local robustness”. Please pay particular attention to reflecting the perspectives brought up by Reviewer 8nCi. I believe that one way forward is to state that the conclusions of the present manuscript point to the incompatibility between $p$-norm based definitions of adversarial robustness and the semantics of tasks which ML is applied to. The authors may find it useful to discuss prior work that identifies this incompatibility.

It would also be beneficial to add the experiment on alpha-beta-crown (https://arxiv.org/abs/2103.06624) requested by Reviewer Br6u.

> Would at least some individuals in TMLR's audience be interested in the findings of this paper?

All 3 reviewers agreed that the paper reports findings that would be interesting to the TMLR audience - despite the current lack of practical solutions to the problem identified by the authors.

## Overall decision

I would like to thank the authors for engaging with reviewers in a discussion following the reviews being posted. I believe that if this discussion is reflected honestly in the manuscript, the resulting paper would be of value to our community. Thus, I encourage the authors to further discuss with reviewers on OpenReview should they have any questions as they revise the manuscript.

---

> ### Author Response · Authors · 2022-11-04
> **Camera-Ready Version Submitted**
>
> We thank the action editor and the reviewers for the detailed discussion and feedback that has helped improve this article significantly. We have uploaded the camera-ready version of the manuscript.
> The primary changes are as follows:
> - We have added a discussion in Section 3 (under paragraph titled “Dichotomy between defending against adversarial examples  and certifying local robustness”) about how the lack of knowledge of $M$ can make the conclusions of this paper non-actionable. In Section 7 (Conclusion), we have also added a list of concrete steps that are possible (even without knowing $M$) to mitigate degradation attacks. Note that Section 5 already discusses the trade-offs of such interventions for mitigating degradation attacks.
>
> - We have added a subsection (Section 3.3) titled “Threat model” that describes our threat model and summarizes the discussion with Reviewer SUEh. The same subsection also has a short discussion about  “the incompatibility between $p$-norm based definitions of adversarial robustness and the semantics of tasks which ML is applied to”.
>
> - We have added a paragraph in Section 3 titled “Dichotomy between defending against adversarial examples and certifying local robustness” motivated by the discussion with Reviewer 8nCi.
>
> - We have added a “Statement of broader impact” in our appendix (Appendix D).
>
> - We think that experiments with alpha-beta-crown will not add to the paper’s message. This is for two reasons - (i) the efficacy of degradation attacks primarily depends on the extent to which the model is $2\epsilon$ robust at the test inputs. Since alpha-beta-crown only presents a certification mechanism but does not present a new algorithm for robust training of neural networks, evaluating attack efficacy with respect to this defense is not expected to yield new insights, (ii) though alpha-beta-crown is a complete certification mechanism, degradation attacks are independent of whether the local robustness check is complete or incomplete.
> For these reasons, we do not include an evaluation with alpha-beta-crown.

---

> > ### Comment · Action_Editors · 2022-11-30
> > **Introduction nitpick**
> >
> > Dear authors,
> >
> > Thank you for your patience, I would like to thank you for the revised manuscript. It addresses all of the expectations set out for the camera ready except one: the introduction could more clearly acknowledge the limitations around the lack of knowledge of M.
> >
> > Would you be able to add 1-2 sentences towards the end of the introduction to mirror the last sentences of the paragraph titled “Dichotomy between defending against adversarial examples and certifying local robustness”? I think this will make the introduction more consistent with the rest of the paper.
> >
> > Once this change is made, I will mark the camera ready as approved.
> >
> > Thank you

---

> > > ### Author Response · Authors · 2022-11-30
> > > **Introduction updated**
> > >
> > > Thank you for the feedback! We have added a couple of sentences at the end of the paragraph that starts with "As we discuss in Section 5 ..." in the introduction. We hope this addresses the concern.